# Connections between Metabolism and Epigenetic Modification in MDSCs

**DOI:** 10.3390/ijms21197356

**Published:** 2020-10-05

**Authors:** Haiyan Dai, Huaxi Xu, Shengjun Wang, Jie Ma

**Affiliations:** Department of Immunology, Jiangsu Key Laboratory of Laboratory Medicine, School of Medicine, Jiangsu University, Zhenjiang 212013, China; 13951408653@163.com (H.D.); xuhx@ujs.edu.cn (H.X.); sjwjs@ujs.edu.cn (S.W.)

**Keywords:** MDSCs, metabolism, epigenetic modification, AMPK, HIF-1α

## Abstract

Myeloid-derived suppressor cells (MDSCs) are major immunosuppressive cells in the tumor microenvironment (TME). During the differentiation and development of MDSCs from myeloid progenitor cells, their functions are also affected by a series of regulatory factors in the TME, such as metabolic reprogramming, epigenetic modification, and cell signaling pathways. Additionally, there is a crosstalk between these regulatory factors. This review mainly introduces the metabolism (especially glucose metabolism) and significant epigenetic modification of MDSCs in the TME, and briefly introduces the connections between metabolism and epigenetic modification in MDSCs, in order to determine the further impact on the immunosuppressive effect of MDSCs, so as to serve as a more effective target for tumor therapy.

## 1. Introduction

Myeloid-derived suppressor cells (MDSCs) are a group of inhibitory cells derived from bone marrow. Suppressive cells of bone marrow origin were first identified and described in cancer patients more than 20 years ago. MDSCs are the precursor cells of dendritic cells (DCs), macrophages and/or granulocytes, and have the ability to significantly suppress the immune cell response [1]. Human MDSCs are defined by the expression of Alpha M-Integrin CD11b and myeloid (CD14 and CD33) or granulocyte/neutrophil (CD15) markers [2]. MDSCs in mice can express CD11b and Gr-1 at the same time. MDSCs mainly consist of two subsets of mononuclear-MDSCs (M-MDSCs) and polymorphonuclear-MDSCs (PMN-MDSCs) (also known as granulocyte-MDSCs (G-MDSCs)) [3]. They are characterized by their immature state and the ability to suppress the immune response. PMN-MDSCs and neutrophils have the same phenotype and morphological characteristics, while M-MDSCs are similar to monocytes and have high plasticity. The differentiation of M-MDSCs into macrophages and DCs is influenced by the tumor microenvironment (TME) [4,5]. Considerable evidence shows that MDSCs negatively regulate the immune response in cancer [6,7] and other diseases such as aging [8] and inflammation [9]. MDSCs can play an immunosuppressive role through a variety of pathways and mechanisms. For example, MDSCs can inhibit lymphocytes by expressing Argininase-1 (Arg-1), inducible nitric oxide synthase (iNOS), reactive oxygen species (ROS), and other substances; induce other tolerant immune cells, such as regulatory T cells(Tregs), regulatory B cells, and tumor-associated macrophages (TAMs); and indirectly inhibit T cells or effector B cells [1]. Meanwhile, in the TME, cancer cells secrete a variety of molecules involved in the aggregation and recruitment of immature bone marrow cells. These molecules include GM-CSF, M-CSF, TGF-β, TNF-α, VEGF, PGE2, COX2, S100A9, S100A8, IL-1β, IL-6, and IL-10 [2,10].

There is increasing evidence that the TME alters myeloid cells by transforming them into powerful immunosuppressive cells [2]. The mechanism of this has not been thoroughly studied. The TME can affect the amplification, differentiation, metabolism, and function of MDSCs through a variety of mechanisms. These mechanisms include metabolic pathways, cellular signaling pathways, and epigenetic modifications. However, an increasing amount of studies have found that these different mechanisms ultimately affect the function of MDSCs by affecting their metabolism to a large extent.

## 2. Metabolism of MDSCs

### 2.1. Glucose Metabolism

The TME is characterized by hypoxia, extracellular adenosine accumulation, elevated lactate levels, and reduced PH [11,12]. In the TME, lack of oxygen and nutrients, and the existence of ROS makes the living conditions very harsh [13]. Cancer cells are known to prefer glycolysis for energy even when oxygen is plentiful, which is known as the Warburg effect [14]. MDSCs, as the most important immunosuppressive cells in the TME, are in the united front with cancer cells in essence, so they have many similarities with cancer cells in terms of metabolism and adaptive survival mechanism. Relevant studies simulated the biological energy metabolism of MDSCs to explore the metabolic state of cells. The study found that the maturation of MDSCs was associated with high glycolytic flux; the pentose phosphate pathway (PPP) and oxidative phosphorylation (OXPHOS) activity were kept at a minimum level to ensure NADPH production and synthesis. Therefore, MDSCs showed heterogeneous metabolic characteristics similar to those of cancer cells. This may be because MDSCs indirectly inhibit the activity of immune cells by competing for carbon sources with immune cells in the TME [15]. However, the metabolism of MDSCs during tumor growth remains to be further studied.

#### 2.1.1. Lactate—An Important Metabolite in MDSCs

At present, an increasing amount of studies have found that during the tumor growth, various metabolic pathways, such as glycolysis, TCA cycle, and glutamine pathway, have undergone great changes. Similar to cancer cells, MDSCs in the TME also have a high level of glycolysis (Figure 1), which contributes to the accumulation of MDSCs in tumor hosts and the immunosuppressive activity of MDSCs [16,17]. Up-regulation of glycolysis can also prevent MDSCs from producing excessive ROS, thus protecting MDSCs from apoptosis. Moreover, glycolytic metabolite phosphoenol pyruvate (PEP), as an important antioxidant, can prevent excessive ROS production, thus contributing to the survival of MDSCs [16]. The inhibition of 2-deoxyglucose (2-DG) on glycolysis has been shown to inhibit the differentiation of MDSCs [18]. As an important product of glycolysis, lactate plays a very important role in the TME [19]. Studies have shown that lactate can stimulate the immunosuppressive properties of MDSCs [20]. The accumulation of lactate in the TME severely limits the functional properties of T cells and NK cells. Pyruvate dehydrogenase (PDH) converts pyruvate to acetyl-CoA, the substrate of the TCA cycle. However, in most cancers, pyruvate dehydrogenase kinase (PDK) is activated and leads to selective inhibition of PDH, resulting in pyruvate being unable to enter the TCA cycle and continuing along the glycolytic pathway to eventually convert to lactate. Dichloroacetic acid (DCA) is an effective inhibitor of PDK, which can promote pyruvate into TCA cycle and thus reduce the lactate level (Figure 1). DCA is considered to be the first discovered metabolic anti-tumor drug, and recent studies have shown that introducing DCA into nanoparticles can specifically target the inhibition of glycolysis of cancer cells without inhibiting the glycolysis of immune cells [21]. Although the potential of DCA has been determined, its mechanism of action on metabolic regulation is not fully understood. In liver cancer treated with Newcastle disease virus (NDV), DCA significantly reduced the release of lactate in cancer cells, activation of signal transducer and activator of transcription 3 (STAT3), up-regulation of indoleamine 2,3-dioxygenase 1 (IDO1), and infiltration of MDSCs. Therefore, DCA can reduce the expression of MDSCs by targeting aerobic glycolysis [22]. In the TME, cancer cells undergo anaerobic glycolysis in order to obtain energy quickly for biosynthesis. This process produces large amounts of lactate, leading to the accumulation of lactate. Studies have shown that aerobic glycolysis is associated with a high MDSC count, low T cell count, and poor human triple negative breast cancer (TNBC) prognosis. Furthermore, the knockout of lactate dehydrogenase A(LDHA), a key enzyme in glycolysis, resulted in the reduction in MDSCs in tumor tissues and the spleen [23]. Meanwhile, exogenous lactate has been shown to increase MDSCs production in vitro cultured mouse bone marrow cells stimulated by GM-CSF and IL-6 [24]. All this evidence indicates that lactate level is closely related to the differentiation and proliferation of MDSCs. The mechanism, however, is not well explained.

#### 2.1.2. The Role of HIF-1α in the Glycolysis of MDSCs

In the TME, MDSCs have extensive glycolysis due to the presence of hypoxia conditions. The glycolytic pathway of MDSCs is regulated by hypoxia inducible factor-1α (HIF-1α), which enhances the immunosuppressive function of MDSCs [25,26,27,28]. HIF-1α was found to be the main cause of the tumor microenvironment effect on the differentiation and function of MDSCs (Figure 1) [25]. As a key regulator of tumor hypoxia response, HIF-1α is a key transcriptional regulatory protein that regulates many key genes [29]. HIF-1α regulates the glucose metabolism and function of MDSCs in a variety of ways. Firstly, HIF-1α activates some glycolytic genes that regulate the production of ATP and ROS [30]. Secondly, HIF-1α induces ectonucleoside triphosphate diphosphohydrolase 2 (ENTPD2/CD39L1) in cancer cells under hypoxia. ENTPD2 can transform extracellular ATP into 5′-AMP, thus preventing the differentiation of MDSCs and promoting the maintenance of MDSCs. Furthermore, inhibition of ENTPD2 can slow tumor growth and improve the efficiency and efficacy of immunocheckpoint inhibitors [31]. Thirdly, HIF-1α activates glucose transporter-1 (Glut-1) and glycolysis-related genes, thereby accelerating the glycolysis rate of MDSCs and providing a large amount of energy, which is conducive to the proliferation, differentiation, immunosuppression of MDSCs, and production of a large amount of lactate into the TME [25]. Fourthly, pyruvate conversion to lactate in the TME is mediated by LDHA, a target of HIF-1α [19]. In addition, HIF-1α significantly altered the function of MDSCs in the TME and promoted the rapid differentiation of MDSCs into TAMs [25,32].

The regulation of HIF-1α on the glucose metabolism of MDSCs is regulated by many other factors. Firstly, SIRT1 (a histone deacetylase dependent on nicotinamide adenine dinucleotide (NAD^+^)) can limit the function and fate of MDSCs by coordinating HIF-1α-dependent glycolysis. SIRT1 deficiency can enhance glycolysis activity of MDSCs [26]. Interestingly, although SIRT1 can directly target HIF-1α to regulate glycolysis, SIRT1 can also act alone to mediate immune responses [30]. Secondly, glucocorticoid receptor (GR) can regulate the function of MDSCs by regulating HIF-1α dependent glycolysis. GR signaling inhibits HIF-1α and HIF-1α -dependent glycolysis in MDSCs, thereby promoting immunosuppressive activity of MDSCs [33]. Thirdly, mTOR signaling activates HIF-1α-mediated transcription under hypoxia conditions, leading to increased glucose, lactate transporter, and glycolytic enzyme expression levels, as well as reduced mitochondrial oxygen consumption, which in turn mediates the transition from OXPHOS to glycolysis [5,34]. The deficiency of mTOR or mTOR-targeted immunosuppressive drug Rapamycin (RPM) can significantly inhibit the glycolytic pathway of MDSCs and reduce the inhibitory activity of MDSCs [18,35]. Interestingly, mTOR signaling can also affect the glucose metabolism of MDSCs in other ways, but this process may depend on specific signaling pathways, such as the phosphatidylinositol 3-kinase (PI3K)-serine-threonine protein kinase (AKT)-mTOR pathway. The PI3K-AKT-mTOR pathway supports cell proliferation through anabolism and has a high glycolysis and glutamine decomposition rate during protein and nucleic acid synthesis [36]. Methionine enkephalin (MENK) enhances the immune response by significantly inhibiting MDSCs and enhancing T cell response. This process is achieved by the reduction in glycolysis and ROS production of MDSCs mediated by PI3K/AKT/mTOR pathway [37].

#### 2.1.3. The Role of AMPK in the Glycolysis of MDSCs

In addition, adenosine 5′-monophosphate (AMP)-activated protein kinase (AMPK) as the regulator of energy metabolism is also of great significance to the metabolic regulation of MDSCs (Figure 1). AMPK inhibits anabolism to reduce energy consumption and activates catabolism to increase ATP production. Increased AMPK signaling was found in MDSCs of tumor-bearing mice and ovarian cancer patients [38]. The activation of AMPK inhibits several major immune-signaling pathways, such as JAK-STAT, NF-κB, C/EBP, CHOP, and HIF-1α pathways, which induce the expansion and activation of MDSCs [20]. AMPK can inhibit glycolysis through the PI3K-AKT-mTOR pathway, promote the development of glycolysis towards OXPHOS, and play an immunosuppressive role [39]. Blocking the activity of AMPK in MDSCs induced by GM-CSF/IL-6 results in decreased cell inhibitory function [40]. In prostate cancer, MDSCs express androgen receptors (AR). Although AR antagonists can enhance the overall survival rate of prostate cancer patients, inhibition of AR signals can inhibit mitochondrial respiration of myeloid cells through the MPC/AMPK signaling pathway, indirectly enhance glycolysis, and further enhance the tumor-promoting function of MDSCs, leading to tumor progression and death of patients [41]. In the TNBC model of mice, tumor glycolysis coordinated the molecular network of AMPK-ULK1, autophagy, and CCAAT/Enhancer-Binding Protein β (C/EBPβ) pathways to affect MDSCs and maintain tumor immunosuppression. Interestingly, C/EBPβ is involved in important life activities, such as cell proliferation and differentiation, tumorigenesis and apoptosis, and inflammatory response, mainly through the regulation of target cell gene transcription. It can promote Arg-1 transcription and thus lead to increased expression of Arg-1. Therefore, restriction of glycolysis not only reduces the number of MDSCs but also weakens the immunosuppression of MDSCs [23]. In addition, GM-CSF derived from cancer cells induces the transcription of AMPKα1 encoding gene Prkaa1 in MDSCs in a STAT5-dependent manner. Inhibition of AMPKα1 not only reduces the immunosuppressive ability of MDSCs but also improves the immunity of anti-tumor CD8^+^ T cells, thereby delaying tumor growth [38].

#### 2.1.4. Type 2 Diabetes Drugs May Inhibit MDSCs by Affecting Glycolysis

It is interesting to note that some type 2 diabetes drugs to treat it that can effectively suppress the immune suppression of MDSCs: this process is dependent on the activation of some special signaling pathways (Figure 2). To some extent, it also indicates that regulating glucose metabolism of MDSCs is crucial to suppress the immune suppression of MDSCs. Phenformin is a biguanide drug used to treat type 2 diabetes. Studies have shown that phenformin has anti-tumor activity in various melanoma models, which enhances the inhibitory effect of the BRAF-MAPK kinase-extracellular signal-regulated kinase pathway and inhibits MDSCs [24].

In addition, metformin, as a drug for the treatment of type 2 diabetes, has attracted much attention in recent years for its anti-tumor effect and its mechanism (Figure 2). Studies have found that metformin can inhibit the aggregation and inhibition of G-MDSCs after mouse colon cancer cell CT-26 transplantation, delay tumor progression, and induce Th1 and CTL responses. In addition, metformin can enhance AMPK phosphorylation, reduce STAT3 phosphorylation, and down-regulate G-MDSC inhibition in vitro [42]. Other studies have found that metformin treatment blocks the inhibitory function of MDSCs in ovarian cancer patients by down-regulating the expression of CD39 and CD73 on M-MDSC and PMN-MDSC subsets and extracellular enzyme activities. At the same time, metformin activates AMPK, thereby inhibiting the expression of HIF-1α, which is a key factor in inducing the expression of CD39/CD73 in MDSCs [43]. Other studies have shown that metformin inhibits MDSCs migration and thus inhibits tumor progression. Metformin inhibits NF-κB expression by increasing AMPK phosphorylation and inducing Dachshund homologue 1 (DACH1) mRNA expression, thereby reducing MDSCs migration and inhibiting CXCL1 secretion in esophageal carcinoma cells and tumor xenografts. Therefore, metformin may play an anti-tumor role by reducing the accumulation of MDSCs in the TME through AMPK/DACH1/CXCL1 axis [44].

In general, metformin inhibits the aggregation and inhibition of MDSCs mainly by activating AMPK-related signaling pathways, and AMPK plays an inseparable role in energy metabolism. Moreover, metformin redirects the metabolism of MDSCs to reduce OXPHOS while increasing glycolysis, thus pushing the microenvironment towards a state of inhibition of certain tumor growth [45]. Therefore, metformin’s inhibitory function on MDSCs is likely to promote their metabolic reprogramming by affecting their glucose metabolism, but the mechanism remains to be further studied.

Therefore, there may be a link between type 2 diabetes and tumor metabolism. By regulating the glucose metabolism of MDSCs may be an important way to treat cancer, and more anti-tumor effects of drugs for type 2 diabetes may be discovered in the future. Although some of these drugs have been used in the clinical treatment of cancer, the mechanism is not fully understood, and the mechanism of their action with glucose metabolism in the TME needs to be further studied. It is believed that with the in-depth study of glycolysis of MDSCs, tumor therapy drugs targeting glycolysis of MDSCs will be more applied in clinical practice.

### 2.2. Lipid Metabolism

Cancer cells usually have a higher rate of denovo fatty acid synthesis to produce cell membrane phospholipids and signaling molecules. A large amount of evidence indicates that lipid metabolism of cancer cells and immune cells in the TME plays an important role in coordinating immunosuppression. Metabolic reprogramming from glycolysis to lipid metabolism is an important regulator of the differentiation and function of different subsets of myeloid cells. Tumor-invasive MDSCs uses fatty acid oxidation (FAO) as its significant source of ATP [46]. Moreover, FAO is the main metabolic fuel for producing inhibitory cytokines, and MDSCs rely on FAO to play an immunosuppressive role [47,48]. After tumor invasion, MDSCs increased fatty acid intake and activated FAO, leading to increased mitochondrial biosynthetic, up-regulated key enzymes of FAO, and increased oxygen consumption rate. FAO as a target is a new way to inhibit the function of MDSCs and enhance the therapeutic effect of various tumors [47]. FAO can also be activated by Stat6, IL4, and other signals that promote MDSC differentiation [49]. Inhibiting FAO can not only block the immunosuppressive function of MDSCs and allow T cells to kill tumor cells but also reduce the production of G-CSF, GM-CSF, and IL6 and promote the anti-tumor response [47]. Interestingly, MDSCs of different subtypes preferred different metabolic modes for energy supply. M-MDSCs preferred FAO rather than glycolysis to provide ATP, while G-MDSCs preferred glycolysis and OXPHO [5]. Although G-MDSCs were the predominant subpopulation of circulating MDSCs in tumor-bearing mice, their immunosuppressive levels were lower than M-MDSCs when assessed on a per-cell basis [50]. In human studies, the number of M-MDSCs, rather than G-MDSCs, was directly associated with the inhibition of T cell activation in vitro [51]. This also indicates that FAO plays a very important role in the immunosuppressive function of MDSCs. Pharmacological inhibition of FAO can block the immunosuppressive function of MDSCs and delay the growth of tumors [47].

Some lipoproteins are also associated with the immunosuppression of MDSCs. Low-density lipoprotein (LDL) is one of the most expressed regulatory factors in PMN-MDSCs, and its receptor oxidized LDL receptor 1 (OLR1) is one of the most overexpressed genes in PMN-MDSCs. OLR1 encoded lectin-type oxidized LDL receptor 1(LOX-1) showed a larger ratio of LOX-1 + in neutrophils of cancer patients, and LOX-1 + neutrophils had the genetic characteristics of PMN-MDSCs, strong immunosuppressive activity, and up-regulated endoplasmic reticulum stress and other biochemical characteristics [52]. In addition, similar to tumor cell metabolism, human tumor infiltrated MDSCs and peripheral blood MDSCs also expressed significant levels of lipid transport proteins. Mouse and human PMN-MDSCs specifically up-regulate fatty acid transporter 2 (FATP2). GM-CSF regulates FATP2 overexpression in PMN-MDSCs by activating STAT5 transcription factor. FATP2-mediated inhibition mainly involves the uptake of arachidonic acid and the synthesis of PGE2. FATP2 deficiency leads to the loss of the inhibitory activity of PMN-MDSCs [53]. Some tumor-derived factors, such as G-CSF and GM-CSF, promote lipid absorption, accumulation, and metabolism in myeloid cells, leading to the induction of MDSC-specific immunosuppressive functions [54]. G-CSF and GM-CSF induce the expression of lipid transporter receptors through signal transduction of STAT3 and STAT5, thus increasing the uptake of high-concentration lipids in the TME and the oxidative metabolism and activating the immunosuppression mechanism. Inhibition of STAT3 or STAT5 signaling or genetic deletion of fatty acid translocation enzyme CD36 inhibited activation of oxidative metabolism and induction of immunosuppressive function in tumor-invasive MDSCs, leading to CD8^+^ T-cell-dependent tumor growth delay [54].

Granulocytes/macrophage progenitor cells (GMP) accumulate during the emergency hematopoietic process driven by cancer and produce MDSCs that express PD-1. In PD-1-deficient myeloid progenitor cells, growth factors that drive emergency hematopoiesis induce an increase in glycolysis, the PPP, and metabolic intermediates in the TCA cycle or lipid metabolism, of which the most significant are elevated cholesterol levels. Cholesterol is essential for the differentiation of inflammatory macrophages and DCs and promotes antigen presentation. Thus, cholesterol production in MDSCs can be inhibited in the presence of PD-1. Therefore, the metabolic reprogramming of emergent myelogenesis and effector myeloid cell differentiation may be the key mechanism of anti-tumor immunity mediated by PD-1 blocking [55]. In addition, liver X receptors (LXRα and LXRβ) are transcription factors of the nuclear hormone receptor family. As a nuclear receptor activated by oxidized sterol, LXRs is involved in the regulation of various physiological activities such as cholesterol metabolism and transport, fat formation, glycogen heterogenesis, and inflammation. LXR agonists have been shown to inhibit tumorigenesis. LXR agonist promotes the up-regulation of its transcription target ApoE, which binds to the LRP8 receptor on MDSCs, thus reducing the survival rate of MDSCs and increasing the CD8^+^ and CD4^+^ T cells activated by tumor invasion, reversing the tumor immune evasion, and promoting anti-tumor immunity [56].

### 2.3. Amino Acid Metabolism

In a hypoxic environment, MDSCs inhibit T cell metabolism by hoarding key amino acids [55]. MDSCs deplete l-cysteine by depletion and sequestration of l-cysteine, resulting in the stagnation of T cell proliferation [2]. Arg-1, which is widely present in MDSCs, consumes a large amount of arginine needed to maintain T cell function [57]. In addition, IDO1 is associated with the expansion, recruitment, and activation of MDSCs in tumors [58]. The main function of IDO1 is to metabolize tryptophan into kynurenine. Tryptophan is one of the main nutrients of lymphocytes. Excessive IDO1 will break down tryptophan in lymphocytes and exhaust it, making lymphocytes become “incompetent state” [59]. Studies have found that the high content of IDO1 in various cancer tissues and their microenvironment is positively correlated with the degree of malignancy and metastasis of cancer. The combination of IDO1 inhibitor and PD-1 antibody can significantly improve the efficiency of cancer immunotherapy [60].

l-glutamine plays a key role in immunosuppression. Glutamine hydrolysis supports the MDSCs maturation process by ensuring the supply of intermediates and energy. In addition, the activity of iNOS can be inhibited under l-glutamine restriction [61]. Recent studies have shown that targeting glutamine metabolism can enhance tumor-specific immunity by regulating MDSCs. The use of small molecule inhibitors of glutamine metabolism not only inhibited tumor growth but also significantly inhibited the generation and recruitment of MDSCs. In addition, inhibition of glutamine metabolism of MDSCs would lead to activation-induced cell death and transformation of MDSCs into inflammatory macrophages. Surprisingly, blocking glutamine metabolism also inhibited the expression of IDO in tumor cells and myeloid cells, leading to a significant decrease in kynurenine. This, in turn, suppresses the development of metastasis and further enhances anti-tumor immunity. Additionally, targeting glutamine metabolism makes it easier for checkpoint blocking drug-resistant tumors to receive immunotherapy [62]. This proves once again that MDSCs and cancer cells have similar metabolic characteristics to a certain extent, and there is a close interaction between the unique metabolism of tumor and the metabolism of inhibitory immune cells.

### 2.4. A New-Type Metabolite of MDSCs

Human MDSCs are characterized by severe metabolic decline and endow CD8^+^ T cells with this impaired metabolic state, thus paralyzing their effector function. A recent study identified the accumulation of dicarbonyl free radical methyl acetaldehyde, produced by semicarbazide-sensitive amine oxidase, leading to the metabolic phenotype of MDSCs and MDSC-mediated paralysis of CD8^+^ T cells. In a mouse cancer model, neutralizing dicarbonyl activity overcomes MDSC-mediated T cell inhibition and, together with checkpoint inhibition, improves the efficacy of cancer immunotherapy. This study identified dicarbonyl methyl acetaldehyde as a marker metabolite of MDSCs, which mediates T cell paralysis and can be used as a target for cancer immunotherapy [63]. This also indicates that the metabolism of MDSCs has not been limited to glucose metabolism, lipid metabolism, and amino acid metabolism. With the in-depth study of MDSCs, an increasing amount of new metabolites and regulatory substances have been discovered gradually.

## 3. Epigenetic Modification of MDSCs

Reprogramming of energy metabolism is thought to be characteristic of cancer. Genomic instability produces genetic diversity and accelerates the acquisition of this trait [64]. Epigenetic modification plays an important role in the regulation of genomes. Epigenetic modification refers to a reversible, heritable change in the function of a gene in the absence of a nuclear DNA sequence change [65]. These changes include modifications of DNA (such as methylation) and various modifications of histones [66] (e.g., acetylation, methylation, phosphorylation, ubiquitination, and recently discovered histone lactylation [67]), RNA interference, etc. Their misregulation may eventually lead to cancer [68]. Because epigenetic modifications are reversible and susceptible to environmental factors, they may be a promising direction for clinical medicine [69]. These epigenetic modifications play an important role in the regulation of cancer cells [70], immunosuppressive cells [71], and tumor-associated fibroblasts [72] in the TME. These cells continuously update and reshape through epigenetic modification in response to a series of signals sent by cancer cells, thereby jointly promoting the occurrence, development, and metastasis of tumors. Many epigenetic modifications have been found in MDSCs, such as DNA methylation, histone methylation, acetylation, ubiquitination, phosphorylation, etc. [73]. In addition to common DNA methylation and histone modification, Non-Coding RNAs also play an important role in epigenetic modification, especially miRNA and siRNA [71], and emerging studies have recently identified the role of miRNAs in MDSCs amplification, development, migration, and function [70]. MDSC-related microRNAs (miR-146a, miR-155, miR-125b, miR-100, let-7e, miR-125A, miR-146b, miR-99b) are associated with the resistance of melanoma patients to immunocheckpoint inhibitor therapy. MDSC-associated miRNA is not only an indicator of MDSC activity in cancer patients but also a potential blood marker for poor immunotherapy outcomes [74]. Moreover, artificial siRNAs targeting different MDSCs genes were used to play anti-cancer effects [71]. Since MDSCs are not a terminally differentiated immune cell population but a heterogeneous mixture of immature myeloid origin cells, they are highly plastic and can be differentiated into DCs or macrophages. Therefore, epigenetic modification plays an important role in the maintenance and activation of MDSCs, and a lot of these epigenetic modifications are carried out through signaling pathways.

### 3.1. DNA Methylation

DNA methylation regulates gene expression by recruiting proteins involved in gene suppression or by inhibiting the binding of transcription factors to DNA [75]. During development, changes in DNA methylation patterns in the genome are the result of a dynamic process involving denovo methylation and demethylation of DNA, which is DNA methyltransferases (DNMTs) dependent [76]. DNA methylation regulates many cellular processes, including embryonic development, transcription, chromatin structure, X chromosome inactivation, genomic imprinting, and chromosome stability [76]. Defects in DNA methylation and its regulation may lead to silencing of tumor suppressor genes or activation of tumor immunosuppression, leading to the occurrence and metastasis of cancer [77,78,79]. Cancers can exhibit global DNA hypomethylation, as well as hypermethylation in genomic regions responsible for tumor suppressor gene expression [73]. DNA methylation also plays a very important role in the proliferation, differentiation, metabolism, and function of MDSCs (Figure 3).

DNA methylation of MDSCs is mainly regulated by DNMTs, and pharmacological inhibition of DNMTs can reduce the accumulation of MDSCs. High demethylation was found in CpG islands of immunosuppressive molecules, such as TGF-DNA1, TIM-3, and promoter regions, which are associated with the immunosuppressive function of MDSCs. MDSCs and DCs are derived from common progenitor cells. Tumor derived factors redirected immune-promoted DCs to tolerant MDSCs. Relevant studies found that DNA methylation in MDSCs was widely demethylated with the increase in DNMT3A level by comparing the DNA methyl groups of MDSCs and DCs. The down-regulation of DNMT3A can eliminate MDSC-specific hypermethylation and cancel the immunosuppressive ability of MDSCs. Consistent with this, DNMT3A was overexpressed and hypermethylated in primary MDSCs isolated from ovarian cancer patients, a process dependent on PGE2 [80]. Studies have shown that TNF-RIP1 mediated necrotizing apoptosis regulates the accumulation of MDSCs. Pharmacological inhibition of DNMT by DNMT inhibitor desitabine (DAC) reduced the accumulation of MDSCs and increased the activation of antigen-specific CTL in tumor-bearing mice. DAC increases cell death by destroying DNA methylation at RIP1-dependent necrotic targets, thereby reducing MDSC accumulation. Whole-genome DNA bisulfite sequencing showed that the promoter was hypermethylated in tumor-induced MDSCs. In addition, IL6 treatment of MDSCs could activate STAT3, increase DNMT1 and DNMT3B expression, and thus improve the survival rate of MDSCs. Thus, this study established a mechanism for MDSC accumulation that MDSCs establish an epigenetic axis of STAT3-DNMT regulated by autocrine IL6 to silence TNFα expression [81]. Immune checkpoint treatment, such as immunotherapy strategies aiming at the programmed cell death ligand 1 (PD-L1)/programmed cell death 1 (PD-1) pathway has achieved significant success, but because of the heterogeneity of tumor and the individual immune system, PD-L1/PD-1 antagonists still show a low response rate in many patients for controlling malignant tumors. An effective response to anti-PD-L1/anti-PD-1 therapy requires the establishment of a complete immune cycle. The damage of immune circulation is an important cause of the failure of immunotherapy, which can be recovered by epigenetic modification [82]. Related studies have investigated the role of DNA methylation in inhibiting molecular transcription in regulatory myelinated inhibitory cells (identified as CD33HLA-DR), and compared with APC. A number of immunocheckpoint (IC), IC ligands, and immunosuppressive molecules associated with the functions of MDSCs were selected, including PD-L1, TIM-3, VISTA, Galectin-9, TGF-β, Arg1, and MMP9. The results showed that the mRNA levels of PD-L1, TIM-3, TGF-β, Arg1, and MMP9 in CD33HLA-DR cells were higher than APCs. Moreover, the CpG islands in the TGF-DNA1, TIM-3, and promoter regions in CD33HLA-DR cells were highly demethylated, suggesting that DNA methylation is one of the key mechanisms to regulate its expression [83].

In addition, different MDSC subsets in the same organism may have different methylation phenomena. Compared with normal tissues, I-MDSCs (immature cells) and PMN-MDSCs were significantly increased in colorectal cancer tissues. Genes associated with DNA methylation mediated transcriptional silencing were up-regulated in tumor invasive I-MDSCs and involved in cell transport and immunosuppression signaling pathways, including Wnt, IL-6, and mitogen-activated protein kinase (MAPK) signaling in I-MDSCs. PMN-MDSCs showed down-regulation of genes associated with DNA methylation and histone deacetylase (HDAC) binding. This suggests that the regulation of gene expression in these MDSC subpopulations is regulated through different epigenetic mechanisms, and the activation of some cell pathways may be realized through epigenetic modifications [84].

### 3.2. Histone Acetylation

Histone acetylation refers to the process in which histone acetyl transferase (HAT) acetylates lysine residues in the histone to activate gene transcription, while HDAC deacetylates the histone to inhibit gene transcription. Histone acetylation uses acetyl-CoA as the substrate to modify the tail of the histone [73]. The effect of histone acetylation on MDSCs is mainly realized by regulating the dynamic balance between HAT and HDAC (Figure 3). TSA is a natural antifungal metabolite produced by Streptomyces that inhibits HDAC and has shown strong HDAC inhibitory activity in various studies [85]. HDAC11 is a key regulator of IL-10 gene expression in myeloid cells. Relevant studies used the mouse transgenic reporting model system in which the expression of EGFP was regulated by the HDAC11 promoter (TG-HDAC11-EGFP), and found that HDAC11 may be a negative regulator of MDSCs expansion/function. Tumor-carrying HDAC11 knockout mice (HDAC11-KO) showed more MDSC inhibition, and the transition from immature myeloid cells to MDSCs required a reduction in HDAC11 expression [86]. In cancer, the normal pathway of monocyte differentiation to macrophages and dendritic cells is changed to preferentially differentiate to PMN-MDSCs. This process is controlled by the epigenetic silencing of the retinoblastoma (Rb) gene controlled by HDAC2. HDAC2 binds directly to the promoter of Rb1, a member of the Rb family that controls cell proliferation and differentiation, and causes Rb1 expression to be silent [87]. l-arginine and l-arginine metabolizing enzymes play important roles in MDSCs. Studies have shown that the induced expression of Arg-1 in differentiated DCs is due to the enhanced recruitment of HDAC4 in the Arg-1 promoter region, which leads to reduced acetylation levels of histone 3 and STAT6 proteins, thus leading to transcriptional activation of Arg-1. There is also a new STAT6 binding site in the Arg-1 promoter, which mediates the regulation of STAT6 and HDAC4 [88]. MDSCs, as precursor cells of DCs, also highly express Arg-1, so this regulatory mechanism is also very important for the differentiation of MDSCs.

In colorectal cancer patients, genes associated with HDAC activation were up-regulated in tumor-invasive I-MDSCs, while genes associated with HAT were down-regulated. Inhibition of HDAC activation or neutralization of IL-6 in colorectal cancer tissues can down-regulate the expression of immunosuppressant and chemotaxis-related genes in myeloid cells, confirming the importance of HDAC activation and IL-6 signaling pathways in the function and chemotaxis of MDSCs [84]. Chromatin modified histone deacetylase inhibitors (HDACi) exhibit anti-inflammatory properties, reflecting their ability to inhibit DC function and enhance Tregs. Exposure of mouse bone marrow cells stimulated by GM-CSF to HDACi significantly amplified M-MDSCs and inhibited the proliferation of Tregs, whose inhibition was dependent on NOS and HO-1. Therefore, HDACi promotes the expansion of MDSCs in vitro and in vivo, suggesting that histone acetylation regulates the differentiation of myeloid cells [89]. The bromine domain (BRD) and the extracellular protein BRD4 bind to acetylated lysine in the histone tail, which lowers acetylation levels and promotes the transcription of pro-inflammatory cytokines. The BET bromine domain inhibitor JQ1 blocks the association between BRD4 and arginase promoter, which promotes immunosuppression of the TME. Combined use of JQ1 and/or PI3K can inhibit BRD4, further inhibit the deacetylation of BRD4, improve the acetylation level of MDSCs, reduce the infiltration of MDSCs, and inhibit the tumor growth in mouse cancer model [90].

The regulation of the acetylation level of MDSCs can also change the immunosuppressive phenotype of MDSCs. The BRD of CBP/EP300 is a key regulator of H3K27 acetylation (H3K27ac) in MDSCs by regulating promoters and enhancers of transoncogenic target genes. In clinical tumor models, the administration of CBP/EP300-BRD inhibitors altered intratumoral MDSCs and tumor growth. Inhibition of CBP/EP300-BRD redirected tumor-associated MDSCs from inhibitory phenotype to inflammatory phenotype by down-regulating the genes associated with the STAT pathway and inhibiting Arg-1 and iNOS [91].

Therefore, due to the role of DNMTs and HDAC inhibition in tumor immune surveillance and escape mechanisms, non-specific inhibition against tumor immune response is currently performed clinically through drug-targeted DNA methylation and histone deacetylation, which has both positive and negative effects. The main negative effect is that these chromatin changes also enhance the immunosuppressive function of MDSCs by up-regulating the expression of checkpoint molecules, impair the function of APC and NK cells, and promote the generation of MDSCs and Tregs, thereby playing an immunosuppressive role [70].

### 3.3. Histone Lactylation

Histone lactylation is a new epigenetic modification proposed recently. It is an epigenetic regulation of gene expression through glycolytic lactate. This modification links epigenetic phenomena with metabolism [67]. Histone lactylation occurs at the amino terminus of the histone lysine, adding lactate groups to the histone lysine and directly stimulating gene transcription. DCA and oxalate inhibit the production of lactate, reduce the level of intracellular lactate, and also reduce the level of histone lactylation. As mentioned above, DCA may reduce MDSCs in a variety of ways, and the reduction in MDSCs may be related to the reduction in histone lactylation level; that is, the high expression of MDSCs in the TME may be closely related to histone lactylation. Further experiments are needed to confirm this conjecture.

## 4. Connections between Metabolism and Epigenetic Modification in MDSCs

There are important bidirectional regulatory mechanisms between metabolic remodeling and the epigenome in cancer. Most chromatin modifying enzymes are required as substrates or cofactors for cell metabolic intermediates. Such metabolites, and the enzymes that normally produce them, can be transferred to the nucleus, directly linking metabolism to nuclear transcription [92]. Therefore, the regulation of MDSCs by epigenetic modification will not only change the differentiation and metabolism of MDSCs themselves but also affect the metabolism of other cells in the TME, enabling these cells to undergo metabolic reprogramming (Figure 4). The epigenome is also sensitive to metabolic states. Metabolites act as cofactors or substrates for several important enzyme reactions related to epigenetic modification and gene regulation [93]. For example, central metabolites are substrates that catalyze the deposition of covalently modified enzymes on histones, DNA, and RNA [94]. Lactate, for example, is an endogenous inhibitor of histone deacetylase, which regulates some genes at the transcriptional level [95]. Therefore, it is particularly important to study the relationship between metabolism and epigenetic modification of MDSCs. At present, epigenetics has been widely studied in the differentiation, proliferation, function, and other aspects of MDSCs, but there are few reports on its effect on metabolism. However, with the in-depth understanding of MDSC metabolism and the important role of cell metabolism on cell function, an increasing amount of studies have begun to focus on the regulation of MDSC metabolism by epigenetic modification.

### 4.1. AMPK and HIF-1α Mediate the Association between Epigenetic Modification and Metabolism

#### 4.1.1. AMPK

The tumorigenesis of MDSCs is mediated in part by the activation of AMPK [96], and this effect was achieved to some extent through the metabolic regulation of MDSCs by AMPK. One of the mechanisms by which AMPK regulates metabolism is histone acetylation. AMPK regulates histone acetylation through a variety of mechanisms. Activation of AMPK can increase the level of acetyl-CoA. Since acetyl-CoA is a substrate for lysine acetyltransferase (KATS), AMPK can affect the activity of KATS by regulating the cell level of acetyl-CoA. In addition, AMPK can activate HDACs by increasing the cell concentration of NAD^+^, a cofactor of HDACs [97]. AMPK also regulates DNA methylation and histone acetylation by phosphorylating epigenetic factors such as DNMT1, retinoblastoma binding protein 7 (RBBP7), and HAT1, thereby promoting mitochondrial biosynthesization and function. AMPK-mediated phosphorylation leads to the activation of HAT1 and inhibition of DNMT1 [98]. Therefore, the regulation of AMPK to regulate the metabolic pathways of MDSCs is achieved to a certain extent through epigenetic regulation, while epigenetic modification is also affected by the level of cell metabolism.

#### 4.1.2. HIF-1α

Epigenetic reprogramming of myeloid cells, also known as training immunity. Induction of aerobic glycolysis by AKT-mTOR-HIF-1α is the metabolic basis of training immunity [99]. Therefore, HIF-1α is also a key regulatory point related to glucose metabolism and epigenetic modification, plays a very important role in metabolic reprogramming of MDSCs, and different cell signaling pathways play a certain role in this process. Since MDSCs exhibit an immature phenotype, the tumor may exhibit either a typically activated phenotype M1 or an alternatively activated phenotype M2. Studies have found that when MDSCs enter the periphery from the bone marrow, SIRT1 (an enzyme responsible for regulating the deacetylation of proteins) deficiency leads to a specific M1 lineage switch, which reduces the inhibitory function of MDSCs and is conducive to the pro-inflammatory M1 phenotype. Differentiation into M1 phenotype requires glycolysis activation of the mTOR-HIF-1 pathway to provide tumor protection. The study identified the nature of the SIRT1-mTOR/HIF-1 glycolysis pathway in determining the differentiation of MDSCs and suggested metabolic reprogramming as a cancer treatment. It is interesting that SIRT1 is a class of NAD^+^ dependent histone deacetylases that are widespread in life [26]. Therefore, phenotypic switching and metabolic reprogramming of MDSCs are likely to be strongly associated with histone deacetylation, and the association needs to be further studied.

### 4.2. Limitations

Although AMPK and HIF-1α can regulate epigenetic modifications such as DNA methylation and histone acetylation, the activation of AMPK and HIF-1α or related signaling pathways can regulate metabolism and function of MDSCs. However, few studies have pointed out that epigenetic modification directly regulates the metabolism of MDSCs. Perhaps in the future, with the in-depth study of MDSCs and the further development of epigenetics, epigenetic modification will make new breakthroughs in regulating the metabolism of MDSCs.

## 5. Conclusions and Prospect

Since the immunosuppressive effect of MDSCs in the TME plays a very important role in the development and metastasis of tumors, it is necessary to study the differentiation, proliferation, metabolism, and function of MDSCs. MDSCs, however, are in a complex environment of tumor microenvironment and highly plastic, so they can be regulated and affected by various factors. In a sense, MDSCs and cancer cells are on the same battle line. Since MDSCs and cancer cells are in the same environment, they have many similar metabolic characteristics, such as aerobic glycolysis, tolerance to high lactate, and glutamine dependence. Therefore, we can proceed from this point and continue to explore the metabolic aspects of MDSCs that are unknown.

With the further development of epigenetics, the emergence of new technologies such as SCNA-SEQ and protein activity integration technology [100], as well as the continuous updating of cell signaling pathways, the current research on MDSCs has become more in-depth, not limited to the study of its classification, differentiation, and expression of specific substances. An increasing amount of studies have been conducted to link MDSCs with epigenetic modifications and signaling pathways to study the effects of epigenetic modifications of MDSCs on their amplification, metabolism, and immunosuppression characteristics and on tumor cells and anti-tumor immune cells. The recently discovered phenomenon of histone lactylation is a major breakthrough in the relationship between cell metabolism and epigenetic modification. However, there are not many studies on the direct impact of MDSCs epigenetic modification on metabolism, and more studies are related to epigenetic modification and cell metabolism through AMPK, HIF-1α, and other signaling pathway-related substances.

Immunosuppression of MDSCs is the key factor of tumor progression and the failure of clinical antitumor immunity treatment; meanwhile, metabolism and epigenetic modification are important factors affecting the development of cancer. Currently, many drugs targeting, respectively, metabolism and epigenetic modification have been used to treat cancer or suppress the development of cancer, such as metformin, DAC, etc. However, due to individual immune differences, heterogeneity, and complexity of cancer, monotherapy is no longer sufficient to meet clinical needs. In addition, multifactorial drug resistance (MDR) impedes the success of cancer treatment, and because of the multifactorial nature of MDR, targeting only one mechanism does not seem to be sufficient to overcome drug resistance. MDR can be reduced by targeting inflammatory processes with immunomodulatory compounds such as mTOR inhibitors, demethylation agents, and low-dose histone deacetylase inhibitors [101]. Only if the relationships between MDSCs and epigenetic modification are thoroughly studied, and relevant anti-tumor drugs can be specifically developed with it as the target, may there be a qualitative leap in tumor therapy. At the same time, excessive regulation should be prevented from having negative effects on APC, NK cells, and other immune cells and damaging their functions. More experiments are needed to confirm this process, such as whether epigenetic modifications can affect mitochondria or other organelles in addition to acting on chromosomes of MDSCs and MDSC-associated immune checkpoint and immunosuppressive molecules. Because many metabolic-related enzymes are present in mitochondria, it is important to link them better to epigenetic modifications. At the same time, in addition to lactate, acetyl-COA, fumaric acid, succinic acid, α-ketoglutaric acid, and other intermediate metabolites in glycolysis and TCA cycle, more metabolites can be put into the study of epigenetic modification and epigenetic modification enzymes. Therefore, it is necessary to establish more comprehensive metabolomic analysis methods and epigenetic analysis methods to examine their regulation of MDSCs with a more comprehensive view, which will be helpful to target the tumor therapy of MDSCs.

## Figures and Tables

**Figure 1 ijms-21-07356-f001:**
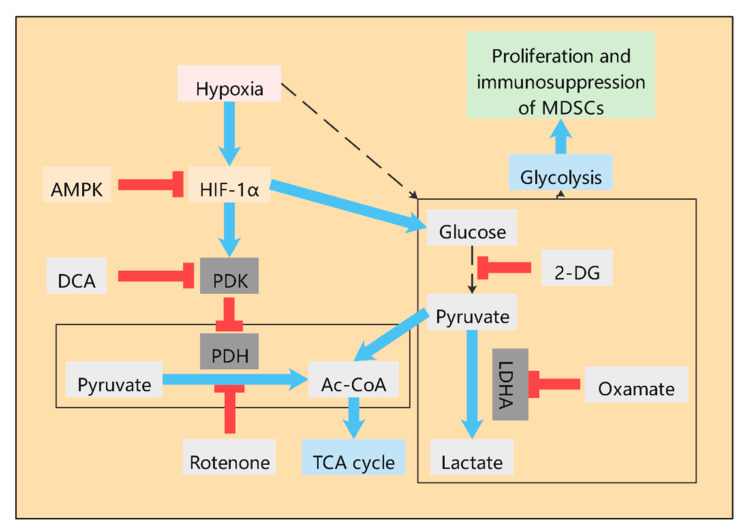
Glucose metabolism in myeloid-derived suppressor cells (MDSCs). High level of glycolysis exists in MDSCs in tumor microenvironment (TME), which contributes to the accumulation of MDSCs in tumor hosts and the play of immunosuppressive activity. As an important product of glycolysis, lactate plays an important role in the TME, which can stimulate the immunosuppressive properties of MDSCs. The non-metabolizable analogue of glucose, 2-deoxyd-glucose (2-DG), inhibits the production of lactate. Oxamate inhibits LDHA directly and thus inhibits the production of lactate. Pyruvate was catalyzed by pyruvate dehydrogenase (PDH) to produce acetyl-CoA and enter the Krebs cycle. However, in most cancers, pyruvate dehydrogenase kinase (PDK) is activated and leads to selective inhibition of PDH, resulting in pyruvate being unable to enter the TCA cycle and continuing along the glycolytic pathway to eventually convert to lactate. Dichloroacetic acid (DCA) is an effective inhibitor of PDK, which can promote pyruvate into the TCA cycle and thus reduce the lactate level. Rotenone directly inhibited PDH to reduce the production of acetyl-CoA, which indirectly promoted the production of lactate. The glycolytic pathway of MDSCs is regulated by HIF-1α and enhances the immunosuppressive function of MDSCs through HIF-1α. AMPK activation can inhibit HIF-1α.

**Figure 2 ijms-21-07356-f002:**
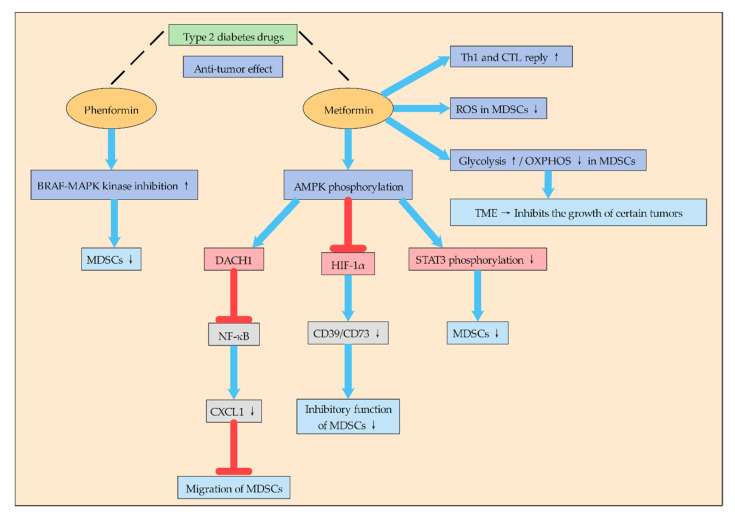
Type 2 diabetes drugs can be used to treat tumors through MDSCs. Type 2 diabetes drugs Phenformin and Metformin can regulate the metabolism of MDSCs through different mechanisms and thus reduce the immunosuppressive function of MDSCs, thus playing an anticancer effect. Phenformin can inhibit MDSCs by enhancing the inhibitory effect of BRAF-MAPK kinase. Metformin can induce the response of Th1 and CTL, reduce ROS in MDSCs, and improve glycolysis and reduce oxidative phosphorylation, thus transforming the TME into a state that inhibits the growth of some tumors. Metformin inhibits MDSCs mainly by activating AMPK phosphorylation and then regulating DACH1, HIF-1α, STAT3, etc.

**Figure 3 ijms-21-07356-f003:**
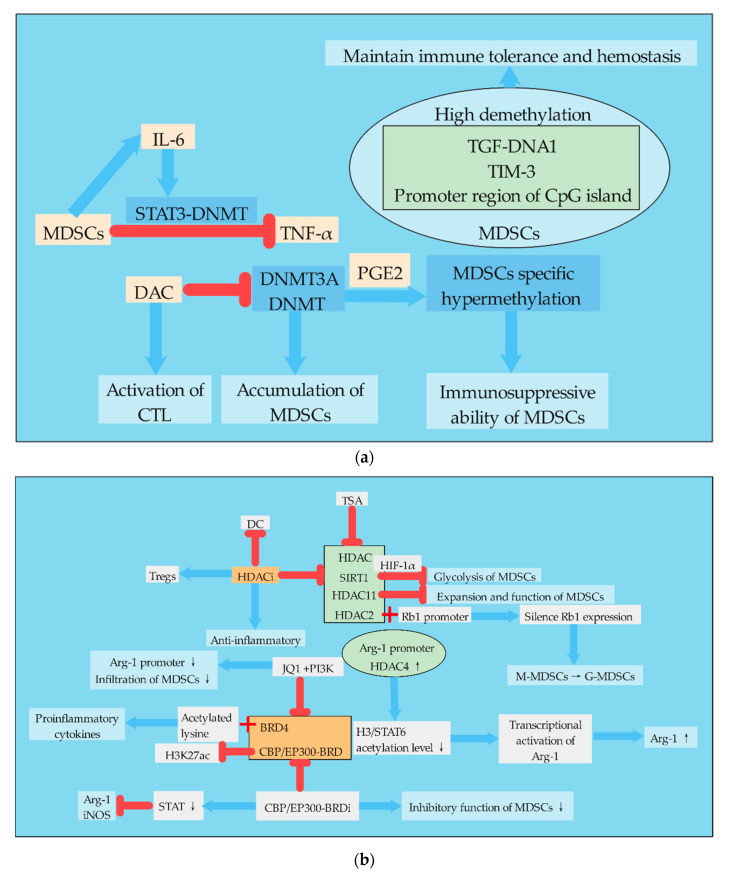
Regulation of epigenetic modification in MDSCs. (**a**). MDSCs can activate the STAT3-DNMT epigenetic axis through autocrine IL-6 to inhibit TNF-α and promote tumor development. High demethylation of TGF-DNA1/TIM-3/CpG island promoter region in MDSCs is of great significance for the maintenance of immune tolerance and hemostasis. DAC can inhibit DNMT and thus the accumulation of MDSCs. DCA can also activate CTL directly. In the presence of PGE2, DNMT3A can promote MDSC-specific hypermethylation. (**b**). Histone acetylation plays an important role in the amplification of MDSCs. The regulation of MDSCs by histone acetylation is mainly realized by the dynamic balance between HATs and HDAC. TSA and HDACi inhibit HDAC. Moreover, HDACi can inhibit DCs and increase the number of Tregs, which has anti-inflammatory effects. HDAC11 can inhibit the amplification and function of MDSCs. The binding of HDAC2 and Rb1 promoter leads to Rb1 silencing and promotes the differentiation of M-MDSCs into G-MDSCs. Arg-1 is a key inhibitory molecule in MDSCs. The increase in HDAC4 in the promoter region of Arg-1 can lead to the decrease in acetylation level of H3/STAT6, thus promoting the transcriptional activation of Arg-1. BRD4 and CBP/EP300-BRD can inhibit histone acetylation and promote the production of proinflammatory factors. CBP/EP300-BRDi can inhibit CBP/EP300-BRD, leading to a decline in the inhibitory function of MDSCs.

**Figure 4 ijms-21-07356-f004:**
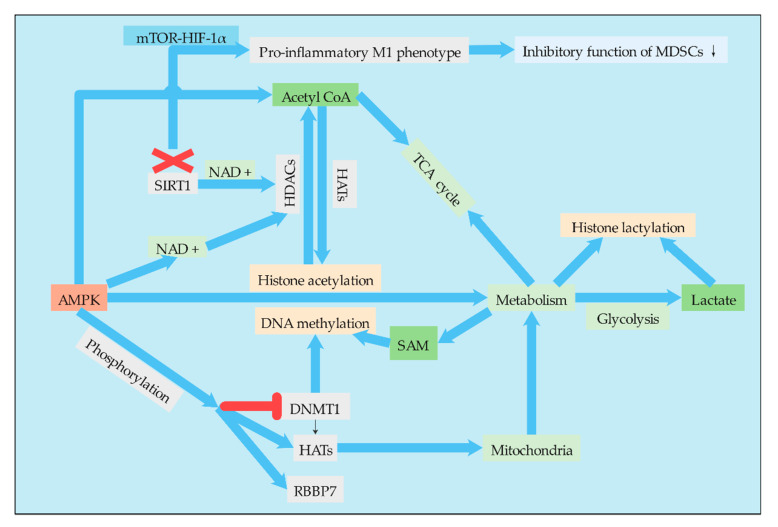
Connections between epigenetic modification and metabolism in MDSCs. Acetyl-CoA, SAM, and lactate, as important substrates and group donors of epigenetic modification, are important pivot substances connecting metabolism and epigenetic modification and have important regulatory effects on epigenetic modification. AMPK, as an energy receptor, can not only regulate metabolism but also regulate some corresponding epigenetic modifications through the regulation of epigenetic modification enzymes such as DNMT, HATs, and HDACs. Moreover, AMPK regulates the metabolism of MDSCs by regulating histone acetylation and DNA methylation.

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
