# Peer review of "Connections between Metabolism and Epigenetic Modification in MDSCs"

_ijms, 2020, doi:10.3390/ijms21197356_

Round 1

Reviewer 1 Report

The review article by Dai et al. describes metabolic reprogramming and significant epigenetic modifications of myeloid-derived suppressor (MDSCs) during tumorigenesis and discusses their reciprocal influences in order to target immunosuppressive functions of MDSCs.

The review addresses a topic currently widely addressed in the scientific literature. Recent and basic literature is cited and processed.

Some minor point could be additionally considered:

- In the section 2.2 “lipid metabolism, the role of CD36-mediated fatty acid uptake in the expansion and regulation of immunosuppressive functions of MDSCs (ref. 47 and 51) should be more extensively addressed, considering the available data. Likewise, some literature about relationship between fatty acid transport protein 2 (FATP2) and MDSCs could be included (e.g. Veglia, F. et al. Fatty acid transport protein 2 reprograms neutrophils in cancer. Nature 2019, 569, 73–78.)

- On the other hand, I think that the authors should refer to the role of oxysterols in MDSCs functions and to the connection between cholesterol metabolism and MDSCs. (e.g. LXR Agonism Depletes MDSCs to Promote Antitumor Immunity. Cancer Discov. 2018, 8, 263; Condamine et al., Lectin-type oxidized LDL receptor-1 distinguishes population of human polymorphonuclear myeloid-derived suppressor cells in cancer patients. Sci. Immunol. 2016, 1.)

- Emerging studies have recently identified the role of miRNAs in the expansion, development, migration and function of MDSCs; in line, artificial siRNAs targeting distinct MDSC genes have been used to exert anti-cancer efficacy. The authors could point out also this point.  

Author Response

Responds to the reviewer’s comments:

Reviewer 1

Thank you for your suggestions. All of your suggestions are very important, and they will be of great guiding significance for my future scientific research. (The reference mentioned in the response have been updated.)

Comment 1: In the section 2.2 “lipid metabolism, the role of CD36-mediated fatty acid uptake in the expansion and regulation of immunosuppressive functions of MDSCs (ref. 47 and 51) should be more extensively addressed, considering the available data. Likewise, some literature about relationship between fatty acid transport protein 2 (FATP2) and MDSCs could be included (e.g. Veglia, F. et al. Fatty acid transport protein 2 reprograms neutrophils in cancer. Nature 2019, 569, 73–78.)

Response: Thank you for your valuable advice. According to your comment, we have supplemented the relevant parts of the article. We have extensively addressed the regulations of CD36-mediated fatty acid on the immunosuppressive functions of MDSCs (ref.54) and have talked about ref.47 and ref.54 in more detail in the section 2.2 “lipid metabolism”. In addition, the relationship between fatty acid transport protein 2 (FATP2) and MDSCs has been included in the section 2.2 “lipid metabolism”. (ref.53)

Comment 2: On the other hand, I think that the authors should refer to the role of oxysterols in MDSCs functions and to the connection between cholesterol metabolism and MDSCs. (e.g. LXR Agonism Depletes MDSCs to Promote Antitumor Immunity. Cancer Discov. 2018, 8, 263; Condamine et al., Lectin-type oxidized LDL receptor-1 distinguishes population of human polymorphonuclear myeloid-derived suppressor cells in cancer patients. Sci. Immunol. 2016, 1.)

Response: Thank you for your valuable and thoughtful comments. We strongly agree with your proposal on adding the role of cholesterol in MDSCs and the relationship between cholesterol metabolism and MDSCs. Cholesterol metabolism is an important part of lipid metabolism and plays an important role in immunosuppression of MDSCs.

According to your suggestion, we have supplemented related contents of cholesterol metabolism in the section “2.2 Lipid metabolism”. (ref.56) In addition, according to your suggestion, we have added related contents about connections between Lectin-type oxidized LDL receptor-1 and PMN-MDSCs in the section “2.2 Lipid metabolism”. (ref.52)

Comment 3: Emerging studies have recently identified the role of miRNAs in the expansion, development, migration and function of MDSCs; in line, artificial siRNAs targeting distinct MDSC genes have been used to exert anti-cancer efficacy. The authors could point out also this point.

Response: Thank you very much. We are very sorry to have ignored the important role of miRNA in the expansion, development, migration and function of MDSCs. Non-coding RNAs also play an important role in epigenetic modification, especially miRNA and siRNA. According to your comment, we have supplemented related contents of miRNAs and siRNAs in the section “3. Epigenetic modification of MDSCs”.

Thanks again for your advice and I hope I can learn more from you.

Reviewer 2 Report

This reviewer manuscript focused on the biological roles of metabolism and epigenetic modification in the context of MDSCs. The three key fields, metabolism, epigenetics and MDSCs, are emerging for their biological and disease relevance. The merit of this manuscript lies in the joint effort to connect the three important fields. This manuscript was organized on the scientific modules of metabolism of MDSCs (glucose, lipid, amino acid, and others) and epigenetic processes of MDSCs (DNA methylation, histone acetylation, histone lactylation) as well as their interaction. The authors did decent work to include the key papers in these fields and outline the current understanding in these fields. The reviewer support publishing this work after addressing the following concerns:

  • Figure 1 is lack of clarification. Basically, two layers of information were embedded in this figure: (a) metabolites and metapolitic enzymes; (b) the upstream biological inputs that define the levels and activities of metapolitic enzymes. The current drawing is very confusing. The authors need to re-draw the figures with listing the key metabiotic reactions including starting materials and products and the enzymes involved. The regulatory mechanism largely acts on the enzyme levels rather than directly on metabolites.
  • For the second part of epigenetic biology, the detailed regulatory mechanisms of DNA methylation and histone acetylation are very vague. More examples are very descriptive and context-dependent with the molecular mechanisms yet to be fully validated. However, in many scenarios, the authors described the mechanisms like a general principle. The reviewer strongly suggested the authors include the context for each discuss case.
  • While the current manuscript made sufficient efforts to review the known work, the authors didn’t give the enough weight to evaluate the current scientific and knowledge gap associated with the topic and how to address these gaps to advance the field.     

Author Response

Responds to the reviewer’s comments:

Reviewer 2

Thank you for your suggestions. All of your suggestions are very important, and they will be of great guiding significance for my future scientific research. (The reference mentioned in the response have been updated.)

Comment 1: Figure 1 is lack of clarification. Basically, two layers of information were embedded in this figure: (a) metabolites and metapolitic enzymes; (b) the upstream biological inputs that define the levels and activities of metapolitic enzymes. The current drawing is very confusing. The authors need to re-draw the figures with listing the key metabiotic reactions including starting materials and products and the enzymes involved. The regulatory mechanism largely acts on the enzyme levels rather than directly on metabolites.

Response: Thank you for your valuable advice. According to your comments, we have redrawn Figure 1, deleted the miscellaneous parts, and listed the key metabolites and key enzymes. At the same time, we have made a detailed description of the figure.

Comment 2: For the second part of epigenetic biology, the detailed regulatory mechanisms of DNA methylation and histone acetylation are very vague. More examples are very descriptive and context-dependent with the molecular mechanisms yet to be fully validated. However, in many scenarios, the authors described the mechanisms like a general principle. The reviewer strongly suggested the authors include the context for each discuss case.

Response: Thank you for your valuable and thoughtful comments. We feel very sorry that we have not fully clarified the detailed regulatory mechanism of DNA methylation and histone acetylation. According to your suggestion, we have discussed the regulation of MDSCs by DNA methylation (ref.80-83) and histone acetylation (ref.86,87,90) in detail in the section “3.1. DNA methylation and 3.2 histone acetylation”.

Comment 3: While the current manuscript made sufficient efforts to review the known work, the authors didn’t give enough weight to evaluate the current scientific and knowledge gap associated with the topic and how to address these gaps to advance the field.

Response: Thank you very much for your instructive comment. We are very sorry to have not well evaluate the current scientific and knowledge gap associated with the topic and how to address these gaps to advance the field. According to your suggestion, we have made corresponding improvements in the section “5. Conclusion and Prospect”.

Thanks again for your advice and I hope I can learn more from you.
